# The Immunological Landscape of M1 and M2 Macrophages and Their Spatial Distribution in Patients with Malignant Pleural Mesothelioma

**DOI:** 10.3390/cancers15215116

**Published:** 2023-10-24

**Authors:** Caddie Laberiano-Fernandez, Camila Machado Baldavira, Juliana Machado-Rugolo, Auriole Tamegnon, Renganayaki Krishna Pandurengan, Alexandre Muxfeldt Ab’Saber, Marcelo Luiz Balancin, Teresa Yae Takagaki, Maria Aparecida Nagai, Vera Luiza Capelozzi, Edwin Roger Parra

**Affiliations:** 1Department of Translational Molecular Pathology, The University of Texas MD Anderson Cancer Center, Houston, TX 77030, USA; cdlaberiano@mdanderson.org (C.L.-F.); atamegnon@mdanderson.org (A.T.); rpandurengan@mdanderson.org (R.K.P.); 2Department of Pathology, Medical School, University of Sao Paulo, Sao Paulo 01246-903, Brazil; ca.mbaldavira@gmail.com (C.M.B.); jurmachado@yahoo.com.br (J.M.-R.); d.saber@uol.com.br (A.M.A.); mlbalancin@me.com (M.L.B.); vera.capelozzi@fm.usp.br (V.L.C.); 3Division of Pneumology, Instituto do Coração (Incor), Medical School, University of Sao Paulo, Sao Paulo 01246-903, Brazil; t.takagaki@hc.fm.usp.br; 4Department of Radiology and Oncology, Medical School, University of Sao Paulo, Sao Paulo 01246-903, Brazil; nagai@usp.br; 5Laboratory of Molecular Genetics, Center for Translational Research in Oncology, Cancer Institute of Sao Paulo, Sao Paulo 01246-903, Brazil

**Keywords:** malignant pleural mesothelioma, transcriptoma, multiplex immunofluorescence, prognosis, in silico analysis

## Abstract

**Simple Summary:**

Identifying biomarkers to guide immunotherapy regimens remains an unmet clinical need in malignant pleural mesothelioma. A potential source of such markers is tumor-associated macrophages (TAMs), which contribute to the immunosuppressive microenvironment of mesothelioma. By examining distinct subsets of pleural macrophages to identify their gene signatures and protein expression, we found that TAMs preferentially contribute to M2a and M2b phenotypes, and M2a, M2b, and M2c more specifically contributed to immune tolerance. *CD206*, *ARG1*, *CD274*, *CD163*, and *MRP8-14* are potential therapeutic targets in this disease.

**Abstract:**

Background: Several tumor-associated macrophages (TAMs) have shown promise as prognosticators in cancer. Our aim was to validate the importance of TAMs in malignant pleural mesothelioma (MPM) using a two-stage design. Methods: We explored The Cancer Genome Atlas (TCGA-MESO) to select immune-relevant macrophage genes in MPM, including M1/M2 markers, as a discovery cohort. This computational cohort was used to create a multiplex immunofluorescence panel. Moreover, a cohort of 68 samples of MPM in paraffin blocks was used to validate the macrophage phenotypes and the co-localization and spatial distribution of these immune cells within the TME and the stromal or tumor compartments. Results: The discovery cohort revealed six immune-relevant macrophage genes (*CD68*, *CD86*, *CD163*, *CD206*, *ARG1*, *CD274*), and complementary genes were differentially expressed by M1 and M2 phenotypes with distinct roles in the tumor microenvironment and were associated with the prognosis. In addition, immune-suppressed MPMs with increased enrichment of CD68, CD86, and CD163 genes and high densities of M2 macrophages expressing CD163 and CD206 proteins were associated with worse overall survival (OS). Interestingly, below-median distances from malignant cells to specific M2a and M2c macrophages were associated with worse OS, suggesting an M2 macrophage-driven suppressive component in these tumors. Conclusions: The interactions between TAMs in situ and, particularly, CD206^+^ macrophages are highly relevant to patient outcomes. High-resolution technology is important for identifying the roles of macrophage populations in tissue specimens and identifying potential therapeutic candidates in MPM.

## 1. Introduction

Malignant pleural mesothelioma (MPM) is a biologically aggressive, lethal malignancy related to asbestos exposure [1,2]. After a prolonged latency following initial exposure, MPM progresses rapidly without effective treatment [3]. Currently, the standard of care for MPM is chemotherapy based on a combination of pemetrexed and cisplatin, followed by radical surgery [4]; however, the response rate to chemotherapy is around 14% [5], and the median overall survival (OS) of patients during chemotherapy is less than 7 months [5]. Predictive biomarkers to guide therapy selection are still lacking [6]. Considering these therapeutic limitations, further studies using technological information gathered from the tumor tissue are necessary to provide opportunities to uncover novel treatment strategies [7]. Another approach that may improve patient survival is early diagnosis based on reliable biomarkers.

The mechanisms of asbestos-induced carcinogenesis have been extensively studied [8,9]. Mesothelial cell (MC) apoptosis is specifically vulnerable to asbestos fiber cytotoxicity [10]. The bio-persistence of asbestos fibers in the milieu induces the escape of MC and macrophages from apoptosis, promoting the oncogenic transformation [11]. Asbestos-driven mesothelial cell transformation and survival are mediated through autophagic pathways [12], and recent reports document a link between HMGB1-driven autophagy and an “inflamed” tumor microenvironment (TME) in asbestos-induced carcinogenesis [13].

Macrophages are the first line of defense against pleural membrane injury because of their exceptional phagocytic capacity. Furthermore, various external signs can induce heterogeneous macrophage subset populations that can be basically characterized as typically activated proinflammatory macrophages (M1) or otherwise activated pro-remodeling macrophages (M2) [14,15]. Similarly, tumor-associated macrophages (TAMs) play a significant role in cancer because they contribute to intimal lining layer hyperplasia [16] and are the leading producers of critical inflammatory mediators that facilitate cancer cell motility and disease progression and form an immunosuppressive TME [17]. The inflamed TME is a significant obstacle to effective tumor-specific T cell cancer response. As the principal residents of the TME [18], TAMs encompass a hugely varied population of different phenotypes, transcription, and functions [19,20] capable of maintaining tumor growth through the secretion of growth factors, matrix degradation enzymes, and proangiogenic factors [21,22,23]. TAMs also suppress immune cells by producing anti-inflammatory cytokines, enzymes that deplete amino acids essential for T cell function (e.g., arginase-1), or by expressing inhibitory immune checkpoint ligands (e.g., PD-L1) [22]. Clinically, TAM density is associated with poor survival in most solid tumors [24]. Macrophages also play a pivotal role in developing effective immunotherapy in some experimental mouse models [25], and the preferential accumulation of proinflammatory M1 TAMs is associated with prolonged survival in patients suffering from several solid tumors [26]. TAMs are the major component of myeloid cell infiltration differentiated from circulating monocytes and display a variety of macrophage subset phenotypes that can be detected by the in situ expression of cell membrane markers, including CD68, CD163, CD206, arginase-1, and MPR8-14 [27,28,29,30]. Immunohistochemical markers used to identify M1 and M2 TAMs are the base of TAM evaluation. M1 macrophages highly express HLA-DR [31], MPR8-14 [30], and inducible nitric oxide synthase (iNOS) in inflammatory tissue [32]. The phosphorylated form of STAT1 (pSTAT1) is a transcription factor that promotes M1-associated functions [33]. Therefore, the immune markers for M1 TAMs in human tissue are HLA-DR, iNOS, MPR8-14, and pSTAT1.

In contrast, the immune markers for M2 TAMs are CD206, CD204, arginase-1, CD163, assignable to the high expression of the mannose receptor-1 (CD206) and macrophage scavenger receptors (CD204 and CD163) by the M2 TAMs. Arginase-1 is a key effector and marker of M2a macrophages and myeloid derived suppressor cells that are major mediators of T cell suppression [30]. Hitherto, M1 macrophages have been investigated mainly for their tumoricidal properties acquired by the secretion of proinflammatory cytokines, including TNF, and activation of a Th1 response [34]. In contrast, the role of M2 macrophages has primarily been investigated in the setting of cancer progression mediated by the secreted growth factors [35]. This knowledge has encouraged the development of various therapeutic agents and clinical trials to target them [21].

The approach of targeting TAMs promises improved cancer outcomes in early clinical results, but response rates are generally low [36]. Nevertheless, specific subsets of TAMs have biological functions, including cytotoxicity and antigen presentation, that are vital at tumor sites [37]. For all these reasons, TAMs have been a promising target for immunotherapeutic strategies, and several approaches have been designed to target crucial aspects of macrophage biology to limit tumor progression, which are currently being explored in clinical trials (NCT02395679 and NCT05765084), as shown in Appendix A [22].

Several immune-relevant gene signatures are associated with the prognostic and predictive value of therapeutic drugs, and the expression of immune-related genes in the tumor could propose the quality and amount of immune infiltration in the tumor microenvironment (TME). Since MPM is an immunogenic tumor, a detailed description of the immune-relevant genes and immune infiltration landscape may result in the promised biomarkers for targeted immunotherapy. Some immune-relevant gene signatures have been described in preclinical models [38,39] and have proven to improve the prognosis of patients with MPM.

The present study first defined the genomic and transcriptomic signatures associated with M1/M2 macrophage differentiation in 87 malignant mesothelioma cases using The Cancer Genome Atlas database (TCGA) as a computational exploratory/discovery cohort. Then, samples from 68 MPM patients were used as a validation cohort to study macrophage subsets based on eight markers at a protein level via a multiplex immunofluorescence panel. Finally, we explored the quantitative and spatial distribution of macrophages and the relationship between the M1/M2 signature and outcomes in the validation cohort.

## 2. Materials and Methods

### 2.1. Computational Exploratory Analysis

#### Exploratory Cohort and Immune Gene Data Collection

The UALCAN platform (http://ualcan.path.uab.edu/ (accessed on 24 April 2023)) [40,41] was used to analyze data from the exploratory cohort consisting of the MPM samples (N = 87) in TCGA (mesothelioma project, Pan-Cancer Atlas) to investigate the relative mRNA expression of immune-relevant genes. GEO2R (http://www.ncbi.nlm.nih.gov/geo/geo2r/ (accessed on 24 April 2023)) was used to screen differential expression genes (DEGs) in MPM based on the analysis of variance or t-test. Then, the fold-change (FC) in the gene expression was calculated with a threshold criterion of |log2FC| 1, and the adj. *p* < 0.05 was set for DEGs selection, resulting in an output of 18 immune genes. Posteriorly, these 18 immune genes of interest (*CD68*, *CD86*, *CD163*, *MRC1*, *ARG1*, *S100A8*, *S100A9*, *CD274*, *NOS2*, *IL1B*, *IL6*, *IL12B*, *CCR7*, *INHBA*, *TNF*, *CHI3L1*, *KCNH6*, *FN1*) were normalized to transcripts per million reads. Using the cBioPortal of Cancer Genomics (https://www.cbioportal.org (accessed on 24 April 2023)) [42,43], we obtained the mRNA expression levels for the markers of interest and mutations, as well as the clinicopathologic characteristics of the patients (Appendix A).

The UALCAN platform was then used to obtain expression profiles associated with the clinicopathologic characteristics, including histologic type, sex, and pathologic stage. The expression levels and characteristics were compared by the platform itself using Student’s t-test. UALCAN also was used to determine the predictive value of these genes. The platform automatically classified patient samples according to the expression of each gene (high vs. low/medium), defining high expression as greater than the third quartile. The significance of the survival impact of gene expression was measured by a log-rank test, with the *p*-values provided.

To explore the functional interactions between the evaluated proteins, the STRING platform was consulted to map the protein–protein interaction (PPI) network and to identify the signaling pathways involved in these interactions [44,45]. Metascape v3.5.20230501 [46] was also used to elucidate the functions and biological processes potentially involved in the enrichment of the genes corresponding to the proteins of interest.

### 2.2. Validation Cohort Analysis

#### 2.2.1. Sample Collection

The validation cohort was composed of 68 tissue specimens, including larger biopsies (N = 7) and surgical resection (N = 67) from patients with MPM diagnosed at the Hospital das Clínicas complex of the Faculty of Medicine of the University of São Paulo. Formalin-fixed paraffin-embedded (FFPE) blocks and hematoxylin–eosin-stained slides were obtained from files at the institutions after informed consent was obtained from all the study participants under protocols approved by the University of Sao Paulo Medical School Institutional Review Board (#2,394,571) and in accordance with the Material Transfer Agreement protocol 2021-0671 of The University of Texas MD Anderson Cancer Center. Two pathologists experienced in thoracic tumors reviewed the slides and classified them using the World Health Organization (WHO) 2021 guidelines [47]. Histologically, the MPM cases were classified by their predominant tumor cell component as epithelioid or sarcomatoid. Because our cases included tumors diagnosed via biopsies, we decided not to include biphasic subtypes by limitation of tissue sample to differentiate between desmoplasia and sarcomatoid components. Tissue microarrays (TMA) were constructed from the tissue specimens using triplicate 2.0 mm diameter cores from the FFPE representative tumor blocks. We collected the clinicopathologic information of the patients, such as their demographic data, age, sex, asbestos exposure, and pathologic stage. Follow-up information for determining the OS rates was also retrieved from the medical records. Table 1 summarizes these clinicopathologic characteristics. 

#### 2.2.2. Multiplex Fluorescence Staining and Analysis

Multiplex immunofluorescence (mIF) staining was performed using methods similar to those previously described and validated [47]. Briefly, formalin-fixed paraffin-embedded TMA sections 4 µm thick from 68 cases of MPM were stained using a macrophage panel against cytokeratin (CK, clone AE1/AE3, dilution 1:25, Dako, Santa Clara, CA, USA), CD68 (clone PG-M1, dilution 1:25, Abcam, Waltham, MA, USA), CD163 (clone 10D6, dilution 1:100, Leica Biosystems, Vista, CA, USA), CD206 (clone PA5-83759, dilution 1:100, Thermo Fisher Scientific, Waltham, MA, USA), Arg-1 (clone D4E3M, dilution 1:100, Cell Signaling Technology, Danvers, MA, USA), MPR8 and MRP14 (MRP8-14, clone 27E10, dilution 1:50, Abcam), CD86 (clone E2G8P, dilution 1:100, Cell Signaling Technology), and PD-L1 (clone E1L3N, dilution 1:100, Cell Signaling Technology). All the markers were stained in sequence using their respective fluorophore contained in the Opal 7 IHC kit (catalog #NEL797001KT; Akoya Biosciences, Marlborough, MA, USA), with the addition of Opal 480 fluorophore (catalog #FP1500001KT; Akoya Biosciences) and Opal 780 fluorophore (catalog #FP1501001KT; Akoya Biosciences) (Appendix A). Positive controls (human reactive tonsils and acute tuberculous pneumonia) and negative controls (human reactive tonsils and acute tuberculous pneumonia including the antibodies but without any fluorophores) were included in the staining run [47]. Appendix A shows the representative individual marker expression from the controls used during the staining.

The stained slides were scanned using the multispectral microscope PhenoImager HT 1.0.13 (formerly Vectra Polaris 1.0.13) system (Akoya Biosciences) under fluorescence conditions at low magnification (10×). Then, each core was viewed at high magnification (20×). Each core from the TMAs was analyzed using the inForm 2.4.0 digital image analysis software (Akoya Biosciences). Marker co-localization was used to identify the possible combinations of cell phenotypes from this mIF panel (Appendix A). The individual cores were divided into the tumor compartment, related to the tumor nests, and defined as clusters of tumor cells surrounded by stroma and the stromal compartment, the tissue between tumor nets. The densities of each cell phenotype were quantified in each compartment, and the final data were expressed as the number of cells/mm^2^ by compartment and total compartment, including the tumor and stroma compartments. The data were consolidated using R studio 3.5.3 (Phenopter 0.2.2 packet; https://rdrr.io/github/akoyabio/phenoptrReports/f/ (accessed on 18 May 2023), Akoya Biosciences).

#### 2.2.3. Cellular Spatial Distribution Analysis

Using the spatial point pattern distribution of the cell phenotypes relative to the malignant cells [48], we measured the distance from the CK^+^ malignant cells to different cell phenotypes quantified by the panel using a matrix created with each cell’s x and y coordinates in R studio software 3.6.1. We applied the median nearest neighbor function (Phenopter 0.2.2 packet) from the CK^+^ malignant cells to the CD68^+^ macrophages and their different subpopulations to determine where these cell phenotypes were located using the dichotomies described below.

### 2.3. Statistical Analysis

The frequency of clinicopathologic data, the median number of macrophage phenotypes by compartment, and the median distances of the macrophage phenotypes from the tumor cells were placed in tables. The Shapiro test was used to determine whether the considered data were normally distributed or not. The Mann–Whitney U nonparametric test was used to compare the continuous median of macrophage phenotypes, histologic types, and asbestos exposure. The association between the distances and clinicopathologic features was evaluated using the Wilcoxon rank-sum or Kruskal–Wallis test. The Spearman nonparametric test was used to assess the associations of the median number of macrophage phenotypes and spatial distances from the malignant cells. A binary split at the median was used to categorize high and low numbers and the distances of the macrophage phenotypes cells to estimate the Kaplan–Meier survival curve. The log-rank test was utilized to compare the differences in the survival curves between different high and low groups. Values above the median were considered a high number or long distance, and values equal to or below the median were considered a low amount or close distance, respectively. Multivariate Cox proportional hazards models were constructed to estimate the association between the number of cells, cellular distances, and risk of death, controlling for clinicopathologic characteristics. An unadjusted *p*-value of less than 0.05 was considered statistically significant. All the analyses and data visualization were performed in R 3.6.0 and 3.6.1 (released April 2019; https://www.r-project.org (accessed on 18 May 2023)), R studio 3.5.3 (Phenopter 0.2.2 packet), and GraphPad Prism 9.0.0.

## 3. Results

### 3.1. Computational Exploratory/Discovery Analysis

#### 3.1.1. Genomic Landscape of MPM

Some gene- or pathway-level somatic mutations with the highest mutation rate overall in MPM may affect the tumor immune microenvironment. Therefore, in our exploratory cohort, we found that the highest genomic alteration rates seen were BAP1 (25.3%), NF2 (21.8%), TP53 (12.6%), TTN (12.6%), SETD2 (9.2%), and LATS2 (9.2%), as shown in Figure 1A.

#### 3.1.2. Transcriptome Profiles of Immune-Relevant Genes in MPM

The eighteen immune-relevant genes to TAM functioning screened according to the fold-change (FC) in the exploratory cohort included *CD68*, *CD86*, *MRC1*, *S100A8*, *S100A9*, *CD274*, and genes involved in M1 and M2 modulation/activation, such as M1 biomarkers *IL1B*, *IL6*, *NOS2*, *IL12B*, *CCR7*, *INHBA*, *TNF*, *CHI3L1*, and *KCNH6*, and M2 biomarkers, including *ARG1*, *CD163*, *FN1*, and *MRP8-14*. For these eighteen genes, we found mutations in *CD86*, *CD163*, *MRC1*, *S100A8*, *S100A9*, *CD274*, *CHI3L1*, *KCNH6*, and *FN1*, as well as deletions and amplifications (Figure 1B). Overall, these results show that structural changes (inversion, deletion, duplication/amplification, and translocation) are infrequent in the MPM exploratory cohort.

Then, to identify any positive or negative associations, we established the correlations between the expression levels of the 18 genes (Figure 1C). A strong direct correlation was found between the expression levels of *CD68* and *CD86* (r_s_ = 0.770, *p* < 0.001), *CD68* and *CD163* (r_s_ = 0.761, *p* < 0.001), *CD86* and *CD163* (r_s_ = 0.785, *p* < 0.001), *CD163* and *MRC1* (r_s_ = 0.669, *p* < 0.001), *S100A8* and *S100A9* (r_s_ = 0.829, *p* < 0.001), *CCR7* and *TNF* (r_s_ = 0.697, *p* < 0.001), and *INHBA* and *FN1* (r_s_ = 0.771, *p* < 0.001). Many moderate correlations were also observed. An inverse correlation was observed between *S100A9* and *IL12B* (r_s_ = −0.211, *p* = 0.05), *NOS2* and *IL1B* (r_s_ = −0.234, *p* = 0.030), *NOS2* and *CHI3L1* (r_s_ = −0.398, *p* < 0.001), *INHBA* and *CHI3L1* (r_s_ = −0.448, *p* < 0.001), and *CHI3L1* and *FN1* (r_s_ = −0.365, *p* = 0.001).

#### 3.1.3. Analysis of Putative Biological Function of the Immune-Relevant Genes through Pathway Enrichment

We investigated the functional association between proteins based on the genomic association of the 18 identified immune-relevant genes (Figure 2). We first visualized the molecular organization of this network made of differentially connected nodes, in which each node represents a gene that encodes functionally interacting proteins (PPI) (Figure 2A). A cluster analysis of this network using the STRING database showed a significant PPI enrichment functional association (*p* < 1.0 × 10^−16^).

A heat map was constructed to access the functions (Figure 2B) and biological processes (Figure 2C) of the proteins encoded by the selected differentially expressed genes. The main terms observed were “response to lipopolysaccharide”, “inflammatory response”, “amoebiasis”, and “positive regulation of inflammatory response”. When we explored the biological process in which these proteins were involved, we found that the principal enriched terms that were most statistically significant were “biological process involved in interspecies interaction between organisms”, “response to stimulus”, and “immune system process”. Upon consultation of the “Reactome Pathways” (Appendix A), the results for these 18 proteins were “Metal sequestration by antimicrobial proteins”, “CD163 mediating an anti-inflammatory response”, “Interleukin-10 signaling”, “Regulation of TLR by endogenous ligand”, “Interleukin-4 and Interleukin-13 signaling,” “Signaling by interleukins”, “Neutrophil degranulation”, “Immune system”, and “Innate immune system”. Here, the highlight is that these functions and pathways identified work together as antitumor and anti-inflammatory defense mechanisms.

#### 3.1.4. Clinical Associations of the Immune-Relevant Genes

We next explored the relation between the mRNA expression of these 18 genes and the clinicopathologic characteristics of the patients using a TCGA database analysis (Appendix A). We found that *CD274*, *IL12B*, and *INHBA* were significantly overexpressed in tumors from male patients (*p* = 0.0004, *p* = 0.001, and *p* = 0.006, respectively; Appendix A). Most of these genes tended to be upregulated in the sarcomatoid histotype, except for *IL1B*, *CHI3L1*, and *KCNH6* (Appendix A). In addition, the biphasic histotype presented upregulation of *IL1B* and *INHBA* compared to epithelioid (*p* = 0.03, *p* = 0.05, respectively), whereas epithelioid showed upregulation of *CHI3L1* compared to biphasic (*p* = 0.02). Furthermore, *IL6* showed higher expression in the biphasic histotype compared to diffuse malignant (*p* = 0.03). Lastly, *CHI3L1* had higher expression in diffuse malignant compared to biphasic (*p* = 0.006).

The expression also varied with the stage and nodal status. *CD163* and *MRC1* were significantly overexpressed in stage IV compared to stage II (*p* = 0.04, *p* = 0.02, respectively). At the same time, *IL6* was significantly overexpressed in stage III compared to stage II (*p* = 0.03) (Appendix A). We also observed considerable overexpression of *CD274* in the N3 category compared to N0 (*p* = 0.03) and of *CHI3L1* in N0 compared to N1 (*p* = 0.03), as well as N2 compared to N1 (*p* = 0.04) and N3 compared to N2 (*p* = 0.05), while *FN1* overexpression was significantly associated with N3 compared to N0 (*p* = 0.0002), N1 (*p* = 0.0009), and N3 (*p* = 0.0001) (Appendix A). However, it is important to point out that only three cases had N3 staging.

#### 3.1.5. Association between Gene Expression and Survival for MPM

To further evaluate the outcome impacts of these 18 immune-relevant and biologically connected genes, we used Kaplan–Meier curves from UALCAN to rapidly assess the effects of these genes on MPM survival. High *NOS2*, *INHBA*, and *FN1* gene expression levels were associated with worse OS (Appendix A).

### 3.2. Validation Cohort Analysis

We validated the transcriptome and genomic data from the TCGA dataset in our cohort of 68 patients with MPM. Using multiplexed image analysis, we evaluated 8 proteins, including CK, CD86, CD86, MRP8, CD163, CD206, Arg-1, and PD-L1 (Figure 3A) in this cohort (Table 2).

#### 3.2.1. M1 and M2 Macrophage Phenotypes

Among a median density of 4202.36 cells/mm^2^ for all the cell types detected across the entire validation cohort, 23% of the cells were CK^+^ malignant cells (median, 966.71 malignant cells/mm^2^), and among the 3235.65 cells/mm^2^ non-malignant cells, only 1.86% (median, 60.37 macrophages/mm^2^) represented CD68^+^ macrophages. Morphologically, the other non-malignant cells and non-macrophage cells were predominantly fibroblasts, inflammatory cells without markers in the panel to be identified as T cells, B cells, neutrophils, and endothelial cells. Using the seven markers from the mIF panel, we identified 14 macrophage phenotypes. We observed high densities of M2 macrophages expressing CD68^+^CD163^+^MRP8-14^neg^CD86^neg^, CD68^+^CD206^+^MRP8-14^+^CD86^neg^, and CD68^+^CD163^+^CD206^+^MRP8-14^neg^CD86^neg^ (median, 21.27, 1.35, and 1.35 cells/mm^2^, respectively), M2a macrophages expressing CD68^+^CD163^+^Arg-1^neg^MRP8-14^neg^CD86^neg^ (median, 17.14 cells/mm^2^), M2b macrophages expressing CD68^+^CD86^+^MRP8-14^neg^ (median, 1.01 cells/mm^2^), and M2c macrophages expressing CD68^+^CD206^+^Arg-1^neg^MRPP8-14^neg^CD86^neg^ (median, 1.13 cells/mm^2^), suggesting a crucial M2 macrophage-driven suppressive component in these tumors. Other macrophages with different phenotypes for M2a, M2b, and M2c were observed, but at very low densities (Figure 3B and Table 2). Interestingly, we observed more macrophages of various phenotypes overall in the tumor compartment compared with the stroma compartment, but the difference did not reach statistical significance, except for the M2 macrophages with CD68^+^CD163^+^Arg-1^+^MRP8-14^neg^CD86^neg^.

#### 3.2.2. Spatial Distances of Macrophages from Malignant Cells

Using the median nearest neighbor distance from malignant cells belonging to various macrophage phenotypes in MPM, we observed that the overall median distance from the malignant cells to the macrophages (CD68^+^) was 73.52 µm (Table 3). By dichotomizing the distances using this median as a cutoff, we observed that all the macrophages characterized as M1 and M2, according to the markers’ co-expression, were located far from the malignant cells (Table 3). The overall distance of the CD68^+^ cells from the malignant cells was within close range, whereas the distances of specific CD68^+^ cell phenotypes were far from the malignant cells, suggesting that the cells expressing only CD68^+^ are macrophages with dendritic linage. The median distances of the M2 macrophages with CD68^+^CD163^+^MRP8-14^neg^CD86^neg^ (median, 123.72 µm) and M2c macrophages with CD68^+^CD163^+^Arg-1^neg^MRP8-14^neg^CD86^neg^ (median, 123.72 µm) were closer to the malignant cells than other M1 and M2 macrophages (Figure 3C).

#### 3.2.3. Associations of Densities and Distance Metrics of Macrophage Phenotypes with Clinical Variables

Overall, the macrophages were closer to the malignant cells in the non-epithelioid mesothelioma and non-asbestos exposure samples (*p* = 0.084, *p* = 0.006, respectively). Regarding the specific phenotypes, M2 macrophages expressing CD68^+^CD163^+^MRP8-14^neg^CD86^neg^, M2a macrophages with CD68^+^CD163^+^Arg-1^+^MRP8-14^neg^CD86^neg^, and M2c macrophages with CD68^+^CD163^+^Arg-1^neg^MRP8-14^neg^CD86^neg^ were closer to the malignant cells in the non-epithelioid group compared with the epithelioid group (*p* = 0.304, *p* = 0.304, *p* = 0.304, respectively), and the same phenotypes were closer to the malignant cells in the non-asbestos group (*p* = 0.070, *p* = 0.061, *p* = 0.061, respectively) (Table 3).

In addition, in the comparison of densities of the macrophage phenotypes between the subgroups with and without asbestos exposure, we found that M2a macrophages with CD68^+^CD163^+^Arg-1^+^MRP8-14^neg^CD86^neg^ and M2c macrophages with CD68^+^CD163^+^CD206^+^Arg-1^neg^MRP8-14^neg^CD86^neg^ showed significantly higher densities in patients with asbestos exposure than in the non-asbestos exposure group (*p* = 0.006, *p* = 0.008, respectively). Similarly, we compared the cellular spatial distances and clinical variables. The median distances for almost all the macrophage phenotypes were closer to the malignant cells in the non-epithelioid histotype when compared with the epithelioid histotype. Interestingly, when we compared patients with and without asbestos exposure, the M1 CD68^+^MRP8-14^+^CD163n^eg^CD206^neg^Arg-1^neg^ macrophages were closer to the malignant cells in the non-asbestos group compared with those with asbestos exposure. In contrast, most M2 macrophage phenotypes were closer to the malignant cells in patients with asbestos exposure compared to the non-asbestos exposure group (Figure 4). No other correlations between densities or distances and other clinical and pathologic features were observed.

#### 3.2.4. Associations between Densities, Distances, and Patient Outcomes

We next examined whether the macrophage densities or distances from malignant cells were associated with patient outcomes. A univariate analysis of the densities showed that above-median densities of cells expressing CD68^+^ were overall associated with better OS than densities less than or equal to the median (Figure 5). Additionally, above-median densities of M2a macrophages with CD68^+^CD163^+^CD206^+^Arg-1^+^MRP8-14^neg^CD86^neg^ expression were associated with worse OS compared with lower densities. Finally, above-median densities of M2c macrophages with CD68^+^CD163^+^Arg-1^neg^MRP8-14^neg^CD86^neg^ were associated with better OS.

A univariate analysis of the cellular distances showed that below-median distances from malignant cells for M2c macrophages with CD68^+^CD163^+^CD206^+^Arg-1^neg^MRP8-14^neg^CD86^neg^ expression were associated with better OS than higher distances. In contrast, below-median distances of M2c macrophages with CD68^+^CD163^+^CD206^+^Arg-1^+^MRP8-14^neg^CD86^neg^ and M2a macrophages with CD68^+^CD206^+^Arg-1^+^MRP8-14^neg^CD86^neg^ were associated with worse OS than higher distances.

Moreover, the Cox proportional hazards regression model, adjusted for histologic type and asbestos exposure, showed that lower densities of M2 macrophages with CD68^+^CD206^+^MRP8-14^neg^CD86^neg^ were associated with better OS. In contrast, lower densities of M2 macrophages with CD68^+^CD163^+^CD206^+^MRP8-14^neg^CD86^neg^ were associated with worse OS (Appendix A). Furthermore, when we further adjusted our Cox proportional hazards regression model using the overall distance of the CD68^+^ macrophages from the malignant cells, lower densities of M2 macrophages with CD68^+^CD206^+^MRP8-14^neg^CD86^neg^ were still predictive of a better prognosis (HR = 0.135, *p* = 0.030), while lower densities of the M2 macrophages with CD68^+^CD163^+^CD206^+^MRP8-14^neg^CD86^neg^ remained an unfavorable OS factor (HR = 7.079, *p* = 0.016) (Table 4).

## 4. Discussion

In the current study, we used the data from The Cancer Genome Atlas and samples in paraffin blocks from 68 patients to investigate the most important genes related to malignant mesothelioma. In our genetic analysis, the highest genomic alteration rate in MPMs was in *BAP1*, which is frequently mutated in patients with immune-activated MPMs and associated with a favorable outcome [49]. Furthermore, we found mutations in *CD86*, *CD163*, *MRC1*, *S100A8*, *S100A9*, *CD274*, *CHI3L1*, *KCNH6*, and *FN1*, and high *NOS2*, *INHBA*, and *FN1* gene expression levels were associated with worse OS. Using mIF, we identified 14 macrophage phenotypes and found high densities of several M2 phenotypes, namely, CD68^+^CD163^+^MRP8-14^neg^CD86^neg^, CD68^+^CD206^+^MRP8-14^+^CD86^neg^, and CD68^+^CD163^+^CD206^+^MRP8-14^neg^CD86^neg^.

M2a CD68^+^CD206^+^Arg-1^+^MRP8-14^neg^CD86^neg^ and M2c CD68^+^CD163^+^CD206^+^Arg-1^neg^MRP8-14^neg^CD86^neg^ macrophages were significantly closer to the malignant cells in patients with asbestos exposure. Finally, on both univariate and multivariate analyses, the M2 phenotype CD68^+^CD206^+^MRP8-14^neg^CD86^neg^ was associated with better OS, while the M2 phenotype CD68^+^CD163^+^CD206^+^MRP8-14^neg^CD86^neg^ was associated with worse OS.

The exploratory analysis of gene signatures, based on scRNA-seq from TCGA, showed that the highest rates of genomic alteration in MPMs were seen in *BAP1*, *NF2*, *TP53*, *TTN*, *SETD2*, and *LATS2*. *BAP1* alteration has been most commonly found in patients with immune-activated MPMs and has been strongly associated with a favorable outcome for these patients, indicating subtype-specific prognostic value [50]. *LATS2* mutation or inactivation is a positive regulator of mesothelioma proliferation via constitutively activating YAP and Hippo signaling pathways [51]. Considering the differentiation of M2-like TAMs and the importance of specific targeted therapy, we explored specific gene signatures for the M1 and the M2 subpopulations. We found that the structural changes in M2 included *ARG1*, *CD206*, *CD163*, *FN1*, and *MRP8*, confirming the potential value of these markers to identify M2-like TAMs in patients with mesothelioma. As previously reported, a lot of genes identified in MPMs were highly expressed by the macrophage population isolated from the pleural effusion and the tumor [52]. This finding supports that TAMs predominantly originate from monocytes in pleural mesothelioma [53,54]. In addition, the functional enrichment analysis of the correspondent proteins showed at least a partial biological connection.

Our multiplexed image analysis showed a selective increase in the immunosuppressive M2-like subset of TAMs that did not extend to proinflammatory M1-like subsets providing an immunosuppressive tumor microenvironment [55]. These subpopulations of TAMs in the TME suggest support for tumor progression through various mechanisms, including suppression of immune cells by potent chemokines (e.g., MRP8-14, CCL2) [30]. These enzymes deplete amino acids essential for T cell function (e.g., arginase-1) or express inhibitory immune checkpoint ligands (e.g., PD-L1) [56]. Notably, we found a higher density of M2 macrophage populations infiltrating the tumor compartment compared to the stroma compartment; therefore, we speculate a crucial role of M2 macrophages’ suppressive component in erecting barriers to effective immunotherapy in these tumors [57], which may explain the peculiar encased growth of MPM. We also observed that the increased population of M2 was followed by a depletion in M2a, M2b, and M2c, suggesting a reduction of T and B cells in the TME. This finding aligns with a recent publication by Wu and colleagues [39], who demonstrated in an experimental model that systemic macrophage depletion decreased tumor progression and reduced cytokines, chemokines, and growth factors in the pleura/peritoneal cavity. In particular, the finding of CCL2 expression in previous studies supports the possibility that this chemokine is a key mediator in recruiting M2 macrophages to the TME [58]. Interestingly, the accumulation of CD68^+^CD206^+^ M2-like TAMs appears to be influenced at least in part by the presence of soluble factors (e.g., folate receptor β) produced by tumor cells in the TME that favor the polarization and recruitment of M2-like macrophages [55]. In other words, our findings are consistent with the poor lymphocyte infiltration and macrophage predominance in MPMs on histologic examination, as demonstrated in our previous study [59].

TAMs can be functionally reprogrammed to polarized phenotypes when they are activated by cancer factors, TME factors, and drug interferences [23]. Since TAMs consecutively differentiate from monocytes into functional macrophages through several phases, they present heterogeneity and plasticity properties in cancer. In tissue specimens, circulant monocytes differentiate into tissue macrophages through the macrophage colony-stimulating factor (M-CSF) and are prepared by several cytokines, such as interferon-gamma (IFN-γ), interleukin 4 (IL-4), and IL-13 [23]. Subsequently, macrophages modify their functional phenotype in reaction to environmental factors or tumor-derived protein stimulation [60]. In skin melanoma, for example, silencing the CD115 receptor (siCD115) in TAMs induces modulation of the tumor interstitial adjustments (TILs) profile, leading to in vivo growth suppression of B16 melanoma [61]. In the next step of priming, IFN-α, IFN-β, and IFN-γ modulate the secretion of chemokines from TAMs, indicating that these cytokines repolarize TAMs in several skin cancers [62]. TAMs in cutaneous cancer also secrete various chemokines to regulate the tumor microenvironment [27]. In skin Paget disease, the cells release soluble RANKL, improving the secretion of CCL5, CCL17, and CXCL10 from RANK^+^ M2 polarized TAMs [63], suggesting that Paget cells can modulate the microenvironment landscape by the stimulation of TAMs. In cutaneous poorly differentiated squamous cell carcinoma, TAMs un-homogeneously polarized from M1 to M2 [64]. These authors also reported that CD163^+^ TAMs not only express CCL18, an M2 chemokine implicated in remodeling of the TME, but are also colocalized with a phosphorylated signal transducer and an activator of transcription 1, suggesting the heterogeneous activation states of TAMs [64]. Hematopoietic malignancies in the skin contain CD163^+^ TAMs, which produce chemokines that direct specific anatomic sites to form metastases [65].

According to our results and the previous study reported, we infer that monocyte-derived macrophages played a predominant role in the development of mesothelioma. Macrophages were the most critical population to infiltrate the tumor and generated the most CD206^+^ M2-like TAM. We also infer that the higher macrophage expression was potentially associated with a drop in T and B cells. We also infer that CCL2 expression remained high, supporting the possibility that this chemokine is a crucial mediator in recruiting monocytes to the tumor microenvironment [58].

Remarkably, we also found that the distance metrics in the MPMs revealed that mostly CD68^+^CD163^+^MRP8-14^neg^CD86^neg^ and M2c macrophages expressing CD68^+^CD163^+^Arg-1^neg^MRP8-14^neg^CD86^neg^ were present in the tumor mass. In contrast, the other M1 and M2 macrophage phenotypes were found at the tumor–stroma border, indicating the existence of distinct macrophage roles in MPM. Similar findings were demonstrated by Egeblad and colleagues in breast cancer, showing that non-migratory macrophages present within the tumor body were mostly CD68^+^ CD206^neg^ [66]. On the other hand, these authors also showed that macrophages at the tumor–stroma border could be identified as migratory CD68^+^CD206^neg^dextran^neg^ myeloid cells and sessile CD68^+^CD206^+^dextran^+^ M2-type TAMs, representing the existence of distinct macrophage types in breast tumors, as well [66].

In the Kaplan–Meier analysis, we found that OS was associated with the TAM number and their proximity to the tumor cells, highlighting the importance of investigating the distribution of cells. Interestingly, the Cox proportional hazards regression model adjusted for histologic type, asbestos exposure, and overall macrophage distance from the malignant cells showed that factors associated with better OS included low densities of M2 macrophages expressing CD68^+^CD206^+^MRP8-14^neg^CD86^neg^. In contrast, lower densities of M2 macrophages expressing CD68^+^CD163^+^CD206^+^MRP8-14^neg^CD86^neg^ were associated with worse OS. In colorectal cancer, Kou et al. showed that a high expression of CD86^+^ and CD68^+^CD86^+^ TAMs, as well as a low expression of CD163^+^ and CD68^+^CD163^+^ cells, were associated with favorable OS [67]. Although we cannot describe a cell-specific mechanism to explain our observation, signs may be found upon further study about the microenvironmental signatures associated with these TAM populations and the nature of the markers expressed. Functionally, CD68, an endosomal/lysosomal glycoprotein highly expressed by the mononuclear phagocytes, is the receptor for apoptotic cells and may be involved in antigen processing [68]. CD163 is a high-affinity haptoglobin–hemoglobin and HMGB1 scavenger (CD206) receptor and has been found to be upregulated on macrophages polarized by IL-10 [69]. This evidence suggests that the CD68^+^CD163^+^ TAMs may work in clearing dead cells, stroma remodeling, and anti-inflammatory processes. High numbers of this population near the tumor cell may reflect an enhanced immunological response in these particular tumors. In addition, CD206^+^ M2-like TAM was highly expressed in mesothelioma and showed a correlation between high CD206^+^ gene expression and worse PFS in the later stages of the disease. It was recently demonstrated to be an essential target for monocyte-derived TAM in other solid tumors [70,71].

As an important limitation of the current study, our specimens were placed in TMA format, which may induce under- or overrepresentation of the marker levels and spatial distribution owing to tumor heterogeneity and the small number of cases in this cohort.

In summary, our results outline the spatial resolution of macrophage polarization in MPM. We dissected the different microenvironmental gene signatures, which may reflect the interactive process between macrophage populations in situ and recognize the CD206^+^ macrophages to be highly relevant in patient outcomes. Our data demonstrate that the polarization of macrophages within the tumor is present at both macro- and micro-levels owing to the gradient change of different markers. Therefore, we highlight the importance of using high-resolution technology to dissect the roles of macrophage populations in tissue specimens, identify potential therapeutic candidates, and understand the immune landscape of MPM in a large cohort.

## 5. Conclusions

The association between TAMs’ in situ expression, particularly CD206^+^ macrophages, is highly relevant to patient outcomes. High-resolution technology is essential for identifying the roles of macrophage populations in tissue specimens and identifying potential therapeutic candidates in MPM.

## Figures and Tables

**Figure 1 cancers-15-05116-f001:**
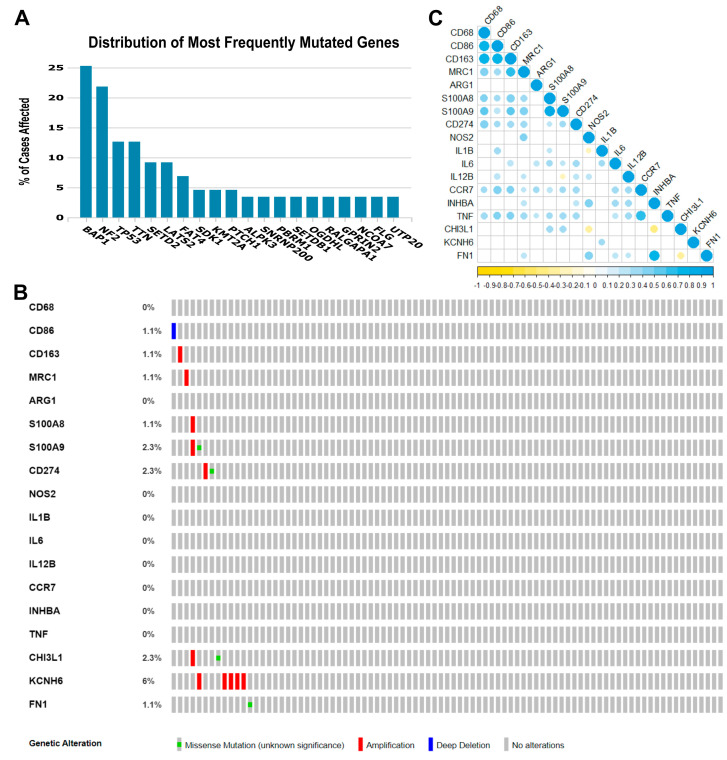
Genomic landscape of malignant pleural mesothelioma and correlations among the 18 immune genes whose expression was obtained from the TCGA Mesothelioma Pan-Cancer Atlas database. (**A**) Distribution of the most frequent mutated genes in The Cancer Genome Atlas (TCGA) database (mesothelioma: Pan-Cancer Atlas). The percentage of cases affected in descending order of frequency for *BAP1*, *NF2*, *TP53*, *TTN*, *SETD2*, and *LATS2* relative to immune-activated gene- or pathway-level somatic mutations. (**B**) Heat map showing immune gene- or pathway-level somatic mutations (missense mutation, amplification, or deep deletion) and their frequency. The images were downloaded from TCGA and the cBioPortal of Cancer Genomic. (**C**) The color gradation represents Spearman’s correlation (blue for positive correlations and yellow for negative). The point size represents the rho value; larger points present rho values closer to |1| and, therefore, stronger correlations.

**Figure 2 cancers-15-05116-f002:**
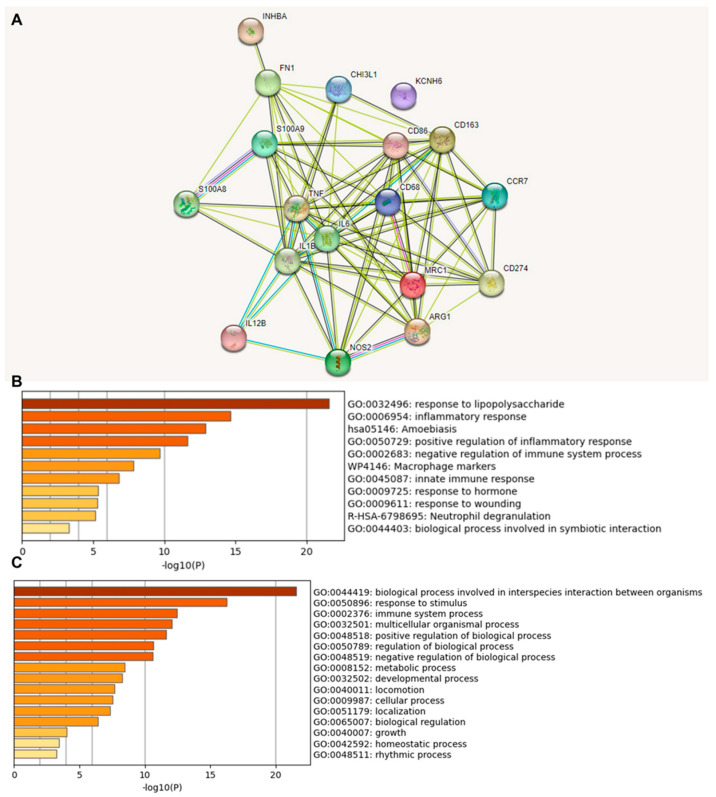
Protein–protein interaction (PPI) network construction and enrichment analysis of 18 TAM genes. (**A**) Cluster analysis of the PPI network using the STRING database for protein interactions. The network included the 18 functional genes with the highest interaction confidence score, namely, *CD68*, *CD86*, *CD163*, *MRC1*, *ARG1*, *S100A8*, *S100A9*, *CD274*, *NOS2*, *IL1B*, *IL6*, *IL12B*, *CCR7*, *INHBA*, *TNF*, *CHI3L1*, *KCNH6*, and *FN1* (*p* < 1.0 × 10^−16^). (**B**) Bar graph of enriched terms across 18 input genes, colored by *p*-values. (**C**) The top-level Gene Ontology biological processes involving the 18 input genes. The images were downloaded from STRING and Metascape after input of 18 immune genes.

**Figure 3 cancers-15-05116-f003:**
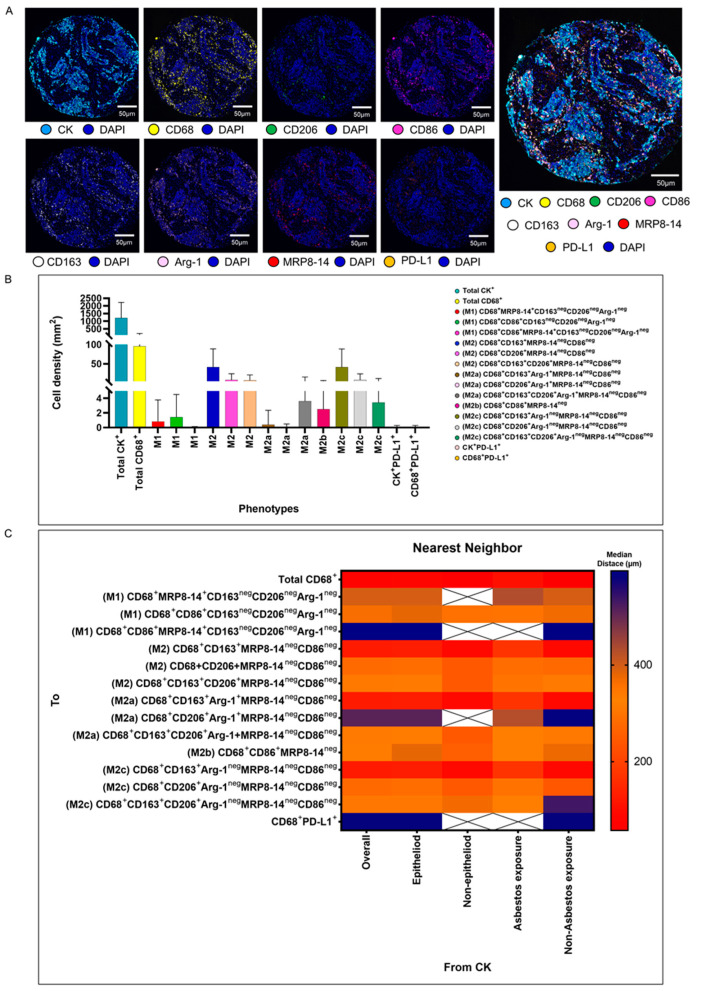
Representative examples of unmixed and mixed multispectral images, bar graphs of macrophage phenotype densities, and heat maps representing nearest neighbor distance analysis. (**A**) Unmixed individual markers, including cytokeratin (CK), CD68, CD163, CD206, Arg1, MRP8-14, CD86, PD-L1, and 4′,6-diamidino-2-phenylindole (DAPI), plus their composite spectral mixing image from multiplex immunofluorescence (mIF; 20× magnification; scale bars represent 50 µm on each image). (**B**) Bar graph of individual densities presented as median values from the macrophage phenotypes observed in the mIF panels. (**C**) Median distance heat map representing 15 macrophage populations near malignant cells (CK^+^), including by histologic type and asbestos exposure. White “X” spaces represent data that are not computable or less than one cell by mm^2^. Data from 68 samples were used. Experiments and quantifications related to the presented results were conducted once. The images were generated using the PhenoImager HT 1.0.13 scanner system and inForm 2.4.0 image analysis software (Akoya Biosciences). The bar graph and the heat map were generated using GraphPad Prism 9.0.0.

**Figure 4 cancers-15-05116-f004:**
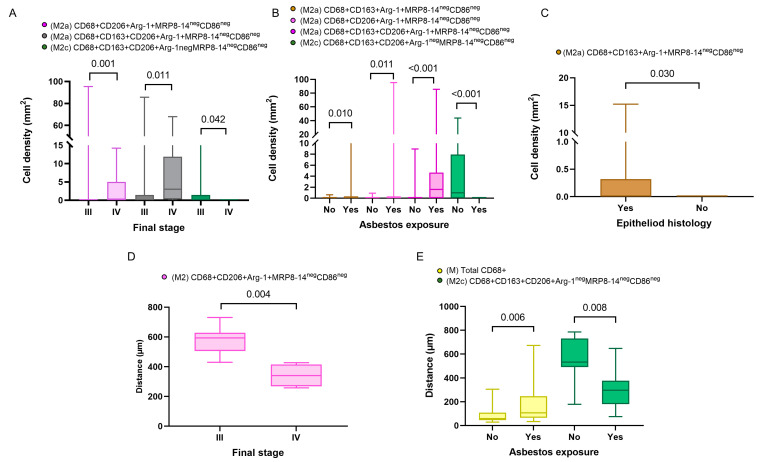
Boxplots showing the significant associations between densities and distances from malignant cells of immune cell populations and clinicopathologic features. Boxplots showing significant associations between densities of different M2 macrophages and clinical variables, including final stage (**A**), asbestos exposure (**B**), and malignant pleural mesothelioma histology (**C**). Boxplots of distances from malignant cells of different macrophage phenotypes and their associations with final stage (**D**) and asbestos exposure (**E**). Data from 68 samples were used. Boxplots show the median (bar), interquartile range (top and bottom), and highest or lowest values. Kruskal–Wallis test was used in (**A**) through (**E**) to compare groups. Boxplots were generated using GraphPad Prism 9.0.0. using unadjusted *p*-values.

**Figure 5 cancers-15-05116-f005:**
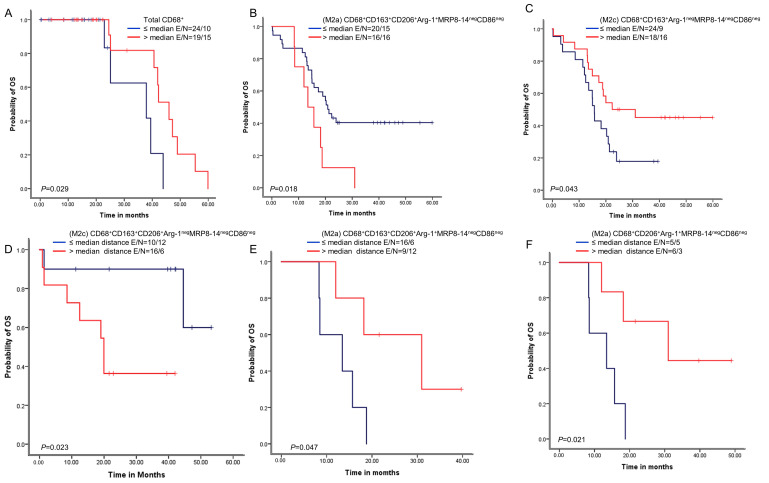
Kaplan–Meier analysis of overall survival (OS) by cellular densities and nearest neighbor distance analysis from malignant cells for various macrophage subpopulations. Red lines indicate high densities (>median) or close (≤median) distances between malignant cells and various macrophages phenotypes, and blue lines indicate low (≤median) densities or long (>median) distances between malignant cells and various macrophages phenotypes. Patients with high total CD68^+^ macrophage density (**A**) and high M2c CD68^+^CD163^+^Arg-1^neg^MRP8-14^neg^CD86^neg^ density (**B**) had better OS than those with lower densities of these macrophage populations. In contrast, patients with high M2a CD68^+^CD163^+^CD206^+^Arg-1^+^MRP8-14^neg^CD86^neg^ density (**C**) had worse OS. Patients with close (≤median) distances from malignant cells to M2c CD68^+^CD163^+^CD206^+^Arg-1^neg^MRP8-14^neg^CD86^neg^ macrophages had better OS (**D**), and patients with close (≤median) distances from malignant cells to (M2a) CD68^+^CD163^+^CD206^+^Arg-1^+^MRP8-14^neg^CD86^neg^ (**E**) and M2a CD68^+^CD206^+^Arg-1^+^MRP8-14^neg^CD86^neg^ (**F**) had worse OS than patients with long distances for these phenotypes. Experiments and quantifications related to the presented results were conducted once. Data from 68 samples were used. Kaplan–Meier curves and log-rank test were used and generated by the R studio software version 3.6.0. with unadjusted *p*-values. E, events; N, censored number.

**Table 1 cancers-15-05116-t001:** Frequency of clinicopathologic characteristics of the 68 patients with malignant pleural mesothelioma whose tumor specimens were included in our study as a discovery cohort.

Characteristics	No. (%)
	n = 68
Age, median (range)	58 years (48–63 years)
Sex	
Female	21 (31)
Male	47 (69)
Asbestos exposure	
No	31 (46)
Yes	37 (54)
Histology types	
Epithelioid	58 (85)
Sarcomatoid	10 (15)
Final stage *	
III	58 (85)
IV	10 (15)
Neo-adjuvant chemotherapy	
Yes	52 (78.6)
No	16 (21.4)
Follow-up	40 months
Overall survival, median (interquartile range)	15 months (5.3–24.8 months)
Vital status	
Dead	44 (62)
Alive	25 (38)

Note: * World Health Organization classification 2021.

**Table 2 cancers-15-05116-t002:** Median number of cell phenotypes by compartment (n = 68).

Phenotype	Total (Range)	Tumor (Range) *	Stroma (Range) *	*p* *
Total CK^+^	995.08 (42–610.94)	1707.04 (121.45–6107.18)	-	-
Total CD68^+^	62.24 (1.83–451.88)	74.20 (1.24–396.92)	54.75 (0.00–645.24)	0.938
(M1) CD68^+^MRP8-14^+^CD163^neg^CD206^neg^Arg-1^neg^	0.00 (0.00–21.53)	0.00 (0.00–37.22)	0.00 (0.00–16.94)	0.254
(M1) CD68^+^CD86^+^CD163^neg^CD206^neg^Arg-1^neg^	0.00 (0.00–20.04)	0.00 (0.00–20.68)	0.00 (0.00–70.93)	0.765
(M1) CD68^+^CD86^+^MRP8-14^+^CD163^neg^CD206^neg^Arg-1^neg^	0.00 (0.00–0.94)	0.00 (0.00–4.14)	0.00 (0.00–0.00)	0.605
(M2) CD68^+^CD163^+^MRP8-14^neg^CD86^neg^	28.33 (0.00–215.25)	29.09 (0.00–153.37)	18.12 (0.00–354.64)	0.657
(M2) CD68^+^CD206^+^MRP8-14^neg^CD86^neg^	1.98 (0.00–69.97)	1.58 (0.00–46.28)	1.14 (0.00–191.07)	0.615
(M2) CD68^+^CD163^+^CD206^+^MRP8-14^neg^CD86^neg^	1.55 (0.00–68.94)	1.22 (0.00–44.92)	1.08 (0.00–155.97)	0.574
(M2a) CD68^+^CD163^+^Arg-1^+^MRP8-14^neg^CD86^neg^	0.00 (0.00–15.20)	0.00 (0.00–16.94)	0.00 (0.00–12.46)	0.010
(M2a) CD68^+^CD206+Arg-1^+^MRP8-14^neg^CD86^neg^	0.00 (0.00–2.26)	0.00 (0.00–2.65)	0.00 (0.00–1.66)	0.089
(M2a) CD68^+^CD163^+^CD206^+^Arg-1^+^MRP8-14^neg^CD86^neg^	0.00 (0.00–67.92)	0.00 (0.00–44.01)	0.00 (0.00–155.97)	0.278
(M2b) CD68^+^CD86^+^MRP8-14^neg^	0.97 (0.00–23.45)	0.48 (0.00–20.68)	0.00 (0.00–82.76)	0.585
(M2c) CD68^+^CD163^+^Arg-1^neg^MRP8-14^neg^CD86^neg^	27.89 (0.00–214.71)	28.31 (0.00–153.37)	18.12 (0.00–354.64)	0.951
(M2c) CD68^+^CD206^+^Arg-1^neg^MRP8-14^neg^CD86^neg^	1.77 (0.00–69.26)	1.56 (0.00–45.37)	1.03 (0.00–191.07)	0.626
(M2c) CD68^+^CD163^+^CD206^+^Arg-1^neg^MRP8-14^neg^CD86^neg^	0.00 (0.00–43.69)	0.00 (0.00–6.28)	0.00 (0.00–151.81)	0.904
CK^+^PD-L1^+^	0.00 (0.00–1.55)	0.00 (0.00–10.05)	-	-
CD68^+^PD-L1^+^	0.00 (0.00–2.48)	0.00 (0.00–0.00)	0.00 (0.00–0.48)	0.608

Note: (M), macrophages; * *p*, correlation between tumor and stroma compartment using Kruskal–Wallis test.

**Table 3 cancers-15-05116-t003:** Median distances from CK^+^ malignant cells to different macrophage phenotypes (n = 68).

	Median Distances from CK^+^ (µm)						
Phenotype	Overall	* Epithelioid	* Non-Epithelioid	* *p*	^#^ Asbestos	^#^ Non-Asbestos	^#^ *p*
Total CD68^+^	73.52	85.49	63.20	0.084	104.34	57.81	0.006
(M1) CD68^+^MRP8-14^+^CD163^neg^CD206^neg^Arg-1^neg^	396.76	396.76	--	--	429.37	396.76	0.972
(M1) CD68^+^CD86^+^CD163^neg^CD206^neg^Arg-1^neg^	363.75	380.13	290.52	0.150	353.23	369.01	0.683
(M1) CD68^+^CD86^+^MRP8-14^+^CD163^neg^CD206^neg^Arg-1^neg^	594.17	594.17	--	--	--	594.17	--
(M2) CD68^+^CD163^+^MRP8-14^neg^CD86^neg^	123.72	127.33	91.79	0.304	170.19	92.69	0.070
(M2) CD68^+^CD206^+^MRP8-14^neg^CD86^neg^	279.00	282.21	243.50	0.565	282.21	275.80	0.805
(M2) CD68^+^CD163^+^CD206^+^MRP8-14^neg^CD86^neg^	305.90	309.01	243.50	0.311	297.79	309.01	0.879
(M2a) CD68^+^CD163^+^Arg-1^+^MRP8-14^neg^CD86^neg^	123.72	127.37	91.79	0.304	170.19	92.69	0.061
(M2a) CD68^+^CD206^+^Arg-1^+^MRP8-14^neg^CD86^neg^	509.77	509.77	--	--	427.64	587.42	0.279
(M2a) CD68^+^CD163^+^CD206^+^Arg-1^+^MRP8-14^neg^CD86^neg^	309.01	314.52	243.50	0.345	317.93	306.11	0.808
(M2b) CD68^+^CD86^+^MRP8-14^neg^	336.39	378.78	255.33	0.028	332.19	373.46	0.851
(M2c) CD68^+^CD163^+^Arg-1^neg^MRP8-14^neg^CD86^neg^	123.72	127.37	91.79	0.304	170.19	92.69	0.061
(M2c) CD68^+^CD206^+^Arg-1^neg^MRP8-14^neg^CD86^neg^	275.80	284.55	243.50	0.593	293.30	240.00	0.481
(M2c) CD68^+^CD163^+^CD206^+^Arg-1^neg^MRP8-14^neg^CD86^neg^	345.72	345.72	367.92	1.000	317.93	533.08	0.008
CD68^+^PD-L1^+^	582.32	582.32	--	--	--	582.32	--

Note: (M), macrophages; * *p*, comparison between epithelioid and non-epithelioid types; ^#^
*p*, comparison between asbestos and non-asbestos exposure using Kruskal–Wallis test.

**Table 4 cancers-15-05116-t004:** Cox proportional hazards regression model of overall survival in patients with malignant pleural mesothelioma (n = 68) comparing low with high densities of different macrophage phenotypes, adjusted for histology type, asbestos exposure, and total CD68^+^ macrophage distance from malignant cells.

Variable	B	SE	Wald	HR	95% CI for Exp(B)	*p*
Histologic type (epithelioid vs. non-epithelioid)	−0.754	0.579	1.695	0.471	0.151–1.464	0.193
Asbestos exposure (yes vs. no)	0.164	0.533	0.095	1.179	0.415–3.351	0.758
Total CD68^+^ (close vs. long distance from malignant cells)	−0.737	0.557	1.747	0.479	0.160–1.427	0.186
Low vs. high densities						
Total CD68^+^	−0.205	1.022	0.040	0.815	0.110–6.037	0.841
(M1) CD68^+^MRP8-14^+^CD163^neg^CD206^neg^Arg-1^neg^	−0.579	0.441	1.721	0.561	0.236–1.331	0.190
(M1) CD68^+^CD86^+^CD163^neg^CD206^neg^Arg-1^neg^	−0.330	0.699	0.223	0.719	0.183–2.830	0.637
(M1) CD68^+^CD86^+^MRP8-14^+^CD163^neg^CD206^neg^Arg-1^neg^	−2.813	1.675	2.821	0.060	0.002–1.600	0.093
(M2) CD68^+^CD163^+^MRP8-14^neg^CD86^neg^	1.336	1.262	1.120	3.802	0.321–45.103	0.290
(M2) CD68^+^CD206^+^MRP8-14^neg^CD86^neg^	−1.999	0.923	4.688	0.135	0.022–0.827	0.030
(M2) CD68^+^CD163^+^CD206^+^MRP8-14^neg^CD86^neg^	1.957	0.812	5.814	7.079	1.442–34.743	0.016
(M2a) CD68^+^CD163^+^Arg-1^+^MRP8-14^neg^CD86^neg^	0.371	0.531	0.488	1.449	0.512–4.105	0.485
(M2a) CD68^+^CD206^+^Arg-1^+^MRP8-14^neg^CD86^neg^	−0.996	0.769	1.679	0.369	0.082–1.666	0.195
(M2a) CD68^+^CD163^+^CD206^+^Arg-1+MRP8-14^neg^CD86^neg^	0.330	0.620	0.284	1.391	0.413–4.692	0.594
(M2b) CD68^+^CD86^+^MRP8-14^neg^	−0.358	0.660	0.294	0.699	0.192–2.550	0.588
(M2c) CD68^+^CD163^+^Arg-1negMRP8-14^neg^CD86^neg^	−0.805	1.249	0.416	0.447	0.039–5.166	0.519
(M2c) CD68^+^CD206^+^Arg-1negMRP8-14^neg^CD86^neg^	−0.273	0.684	0.159	0.761	0.199–2.907	0.690
(M2c) CD68^+^CD163^+^CD206^+^Arg-1^neg^MRP8-14^neg^CD86^neg^	1.364	0.847	2.591	3.910	0.743–20.573	0.108

Note: B, unstandardized regression weight; CD, CI, confidence interval; HR, hazard ratio; (M), macrophages; neg, negative; SE, multiple linear regression; Wald, Wald test.

## Data Availability

The authors declare that the data supporting the findings of this study are available within the manuscript and its Appendix A. Other data related to the current study are available from the corresponding author (E.R.P.) upon academic request and will require the researcher to sign a data access agreement with the University of Sao Paulo Medical School and The University of Texas MD Anderson Cancer Center after approval.

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
