# Peer review of "The Immunological Landscape of M1 and M2 Macrophages and Their Spatial Distribution in Patients with Malignant Pleural Mesothelioma"

_cancers, 2023, doi:10.3390/cancers15215116_

Round 1

Reviewer 1 Report

The authors described the discovery of immunotherapeutically relevant biomarkers on TAM found in the TEM of MPM patients by analyzing gene expression, protein expression, spatial distribution in the TEM, and clinical outcome relevance. Overall, the study is well-researched, enabled by powerful databases, high-resolution techniques, and supported by compelling evidence, so I recommend accepting the manuscript with some revisions needed.

1. Could you specify which specific nonparametric tests were used?

2. An unadjusted P-value of less than 0.05 was used as the threshold for multiple group comparisons. Can you comment on the reason for not using adjustment (e.g., Bonferroni correction) and the potential statistical error?

3. Please justify using the dichotomizing continuous variables method as it can lead to loss of statistical power. Overall, adding more rationales and details in the statistical section could be helpful.

4. There are some minor typos, and please proofread. 

5. Make sure to use high-resolution figures for the final draft.

6. Given the rich observational data, adding more literature in the discussion section related to mechanistic insights could be helpful.

Author Response

Reviewer 1

The authors described the discovery of immunotherapeutically relevant biomarkers on TAM found in the TEM of MPM patients by analyzing gene expression, protein expression, spatial distribution in the TEM, and clinical outcome relevance. Overall, the study is well-researched, enabled by powerful databases, high-resolution techniques, and supported by compelling evidence, so I recommend accepting the manuscript with some revisions needed.

Thank you to the reviewer for their positive comments that allowed us to improve the work to make it more fluent and transparent to justify the study's rationale and minimize major weaknesses in data presentation and interpretation. We revised the manuscript based on these comments and highlighted the modification in the new version in yellow. 

Query 1. Could you specify which specific nonparametric tests were used?

Response. Thank you for the question. In the new version of the manuscript, we re-write the section on statistical analysis, making it more concise and clear for the readers. This is described in Lines 223- 240 

 Query 2. An unadjusted P-value of less than 0.05 was used as the threshold for multiple group comparisons. Can you comment on the reason for not using adjustment (e.g., Bonferroni correction) and the potential statistical error?

Response.  Thank you for the reviewer’s comment. We adjusted the P values using the Benjamini-Hochberg model; unfortunately, our significative p values were drop off. We believe it is because of our small cohort, the use of the TMA instead of the whole section, and the type of tumor we are studying that enters the cold tumor category. These limitations were mentioned in the previous manuscript version; as we said, examining a large cohort is essential to validate our results.

Query 3. Please justify using the dichotomizing continuous variables method as it can lead to loss of statistical power. Overall, adding more rationales and details in the statistical section could be helpful.

Response. We agree with the reviewer. In the present study, we used dichotomization only to estimate the distance of macrophage phenotypes from malignant cells and the Kaplan-Meier survival curves. The remaining analyses were performed with continuous variables data. Attended the reviewer's suggestion in the new version of the manuscript, we re-write the section on statistical analysis. This is described in Lines 223- 240 

Query 4. There are some minor typos, and please proofread.

Response: The manuscript was carefully proofread. Thank you for the very pertinent comment.

Query 5. Make sure to use high-resolution figures for the final draft.

Response. We appreciate this question very much. In the new version of the manuscript, we worked to improve the quality of the figures and submitted individual Figures in high quality according to the journal guidelines. Thank you.

Query 6. Given the rich observational data, adding more literature in the discussion section related to mechanistic insights could be helpful.

 Response. We agree with the reviewer. In the new version of the manuscript, we improved the mechanistic insights in the Discussion section. Thank you for the suggestion. Pleases, check Lines 519-539.

Reviewer 2 Report

It is a quite interesting study. The authors investigated TAMs in MPM, which are most abundant immune infiltrate in mesothelioma. However, the data is not very well presented and organized. There are major comments:

1.      The figures’ resolution is so low that it’s hard to look and read. The images are blurred in main figures. Also, the font size is too small in supplementary figures.

2.      Try to put most of supplementary figures into main figures, which will be more convenient for people to read.

3.      The font size of text in tables is too big, which occupy too much space. The font size should not be bigger than main text.

4.      In figure 1, the authors should show single channel and merged image of one punch of the TMA.

5.      One suggestion for the labelling, put the supplementary table 3 (definition for different subtypes) into main text and use the abbreviations of all the macrophage subtypes in figures or other tables.

6.      In figure 1E, some squares are empty. What does it mean? Is no data available? Better indicate it in figure legend.

7.      The ‘+’ should be superscript, like CD68+. The authors need to correct it in many places (tables and text).

8.      In the introduction part, it will be better that the authors add a brief introduction for different markers of macrophages (18 markers mentioned through the article).

9.      In authors’ patient cohort, the subtypes of MPM include only epithelioid and sarcomatoid types. There are two more subtypes existing, biphasic (30-40%) and desmoplastic type (below 1%). The author should correct it in the text ‘non-epithelioid (sarcomatioid)’, line 148. Is there no biphasic subtype in authors’ own cohort?

10.   In the method part ‘sample collection’, the authors mentioned 8th Edition of the TNM classification for lung cancer. What lung cancer? Line 152. The reference 37 seems not fitting the content.

11.   In the table 1, it’s more meaningful to indicate if the patient receives neoadjuvant chemotherapy, comparing with ‘adjuvant chemotherapy’ status. Because we will know if the tissue was collected from the surgery after chemotherapy.

12.   It will be also very interesting to stain CD8+ T cell markers and evaluate the distance between different subtypes of macrophages and T cells to investigate which subtype of macrophages plays a more important role in immune suppression. 

Author Response

Reviewer 2

It is a quite interesting study. The authors investigated TAMs in MPM, which are most abundant immune infiltrate in mesothelioma. However, the data is not very well presented and organized. There are major comments:

Thank you to the reviewer for their positive comments that allowed us to improve the work to make it more fluent and transparent to justify the study's rationale and minimize major weaknesses in data presentation and interpretation. We revised the manuscript based on these comments and highlighted the modification in the new version in yellow. 

Query 1.      The figures’ resolution is so low that it’s hard to look and read. The images are blurred in main figures. Also, the font size is too small in supplementary figures.

Response. Thank you for the reviewer's comment. In the new version of the manuscript, we worked to improve the images' quality per the journal guidelines and submitted the individual Figures.

Query 2Try to put most of supplementary figures into main figures, which will be more convenient for people to read.

Response. Again, we appreciated this suggestion very much. We attended this request of the reviewer as can be appreciated in the new version of the manuscript.

Query 3The font size of text in tables is too big, which occupy too much space. The font size should not be bigger than main text.

Response. Thank you to the review’s comment. In the new version of the manuscript we modified the font size of text in the tables accordingly the reviewer suggestion.

Query 4.      In figure 1, the authors should show single channel and merged image of one punch of the TMA.

Response. Thank you for the review’s comment. As suggested by the reviewer, in the new version of the manuscript, we replaced the composite image with one representative core showing a single channel and merged image.

Query 5. One suggestion for the labelling, put the supplementary table 3 (definition for different subtypes) into main text and use the abbreviations of all the macrophage subtypes in figures or other tables.

Response. However, the suggestion of the reviewer is good. Maintaining the label of makers in each cell phenotype will be necessary for the readers.

Query 6. In figure 1E, some squares are empty. What does it mean? Is no data available? Better indicate it in figure legend.

Response.  It is an important observation from the reviewer. In the manuscript's new version, we indicate why some squares are empty.

Query 7. The ‘+’ should be superscript, like CD68+. The authors need to correct it in many places (tables and text).

Response. Thank you for the review’s comment. In the new version of the manuscript, as the reviewer suggested, the ‘+’ was superscript.

Query 8. In the introduction part, it will be better that the authors add a brief introduction for different markers of macrophages (18 markers mentioned through the article).

Response. Thank you for the review’s comment. In the new version of the manuscript, we briefly included a paragraph about the immune markers used in this manuscript. Please, review Lines 72-102.

Query 9.  In authors’ patient cohort, the subtypes of MPM include only epithelioid and sarcomatoid types. There are two more subtypes existing, biphasic (30-40%) and desmoplastic type (below 1%). The author should correct it in the text ‘non-epithelioid (sarcomatioid)’, line 148. Is there no biphasic subtype in authors’ own cohort?

Response. We appreciate this question very much. The reason for not including the Biphasic is related to tumor sampling since our cohort has biopsies that do not allow us to differentiate clearly between desmoplasia and sarcomatoid components. Please, check Lines 166- 168 

Query 10.   In the method part ‘sample collection’, the authors mentioned 8th Edition of the TNM classification for lung cancer. What lung cancer? Line 152. The reference 37 seems not fitting the content.

Response. Thank you for highlighting it. In the new version of the manuscript, we corrected this error in the sample collection section.

Query 11.   In the table 1, it’s more meaningful to indicate if the patient receives neoadjuvant chemotherapy, comparing with ‘adjuvant chemotherapy’ status. Because we will know if the tissue was collected from the surgery after chemotherapy.

Response. We appreciate this suggestion very much. However, our patients did not receive adjuvant chemotherapy. Thank you for the very pertinent comment.

Query 12.   It will be also very interesting to stain CD8+ T cell markers and evaluate the distance between different subtypes of macrophages and T cells to investigate which subtype of macrophages plays a more important role in immune suppression.

Response. We appreciate the reviewer’s suggestion. The main objective of this study was to characterize the macrophage population in the malignant pleural mesothelioma. Your suggestion is important, and we will consider it in our subsequent project, which will include a large cohort with the application of a hi-plex panel, including macrophages and lymphocyte markers. Also, we are adding for the reviewer this reference (https://pubmed.ncbi.nlm.nih.gov/33633208/) from a previous study in a small cohort of malignant pleural mesothelioma that studied the distances of lymphocytes, including the CD8. Please, check the line 527-528.

Round 2

Reviewer 2 Report

The quality of figures can still be improved. Authors can try to use Adobe Illustrator (AI) to generate the figures, if possible. That will give high-quality figures.

English is fine. Just check the spelling mistakes.

Author Response

Thank you to the reviewer for their positive comments. We revised the manuscript based on these second comments and highlighted the modification in the new version in yellow. 

Query 1. Supplementary figures are not updated. Figure 1C is the same with Figure S4. Still some space to improve. For instance, supplementary figure 5 can also be put into main figure.

Response: Thank you for your helpful comment. The figures were updated in the new version of the manuscript, and the supplementary figure 5 was included as another main figure.

Query 2. The font sizes are too small in some figures, including supplementary figures (boxplots and survival data). Please increase it. And the quality of the text in figures. It’s hardly visible

Response: Thank you for the reviewer comments. In the new version of the manuscript, we try to increase the font size as much as we can. However, the images were generated by the UALCAN platform, which has limited font size. To minimize this, we increased the quality of the pictures.

Query 3. If possible, can authors also add the introduction for markers MRP8-14, and Arg-1?

Response: Thank you for your reviewer’s comments. In the manuscript's new version, we added the markers MRP8-14 and Arg-1 in the introduction section, as suggested by the reviewer.

Query 4. Line 167, ‘idifferentiate’ should be ‘differentiate’. Please check spelling mistake in the manuscript.

Response: Thank you for your detailed review and for catching the typo. We corrected the typos in the new version of the manuscript.

Query 6. Did authors mean patients didn’t receive neoadjuvant chemotherapy? The authors wrote adjuvant treatment status in the manuscript. Please double check with your clinical doctor if your patients received neoadjuvant chemotherapy or not. Normally patients received neoadjuvant chemotherapy for downstaging before surgery. And the authors’ patient cohort included patients at relatively higher stages, stage III and IV.

Response: The reviewer is entirely correct and thank you for a thorough review. In the manuscript's new version, we correct that error in table 1.